# Median nerve behavior at different wrist positions among older males

Ping Yeap Loh[1], Hiroki Nakashima[1] and Satoshi Muraki[2]

[1] Department of Human Science, Graduate School of Design, Kyushu University, Minami-ku, Fukuoka, Japan
[2] Department of Human Science, Faculty of Design, Kyushu University, Minami-ku, Fukuoka, Japan

## ABSTRACT

The effect of wrist flexion-extension on the median nerve appearance, namely the cross-sectional area (MNCSA) and the longitudinal (D1) and vertical (D2) diameters, was investigated among older adults ($N = 34$). Ultrasound examination was conducted to examine the median nerve at different wrist angles (neutral; and 15°, 30°, and 45° extension and flexion), in both the dominant and nondominant hand. Median nerve behavior were significantly associated with wrist angle changes. The MNCSA at wrist flexion and extension were significantly smaller ($P < .001$) compared with the neutral position in both the dominant and nondominant hand. The D1 and D2 were significantly reduced at flexion ($P < .001$) and extension ($P < .001$), respectively, in both the dominant and nondominant hand. Our results suggest that a larger flexion-extension angle causes higher compression stress on the median nerve, leading to increased deformation of the MNCSA, D1, and D2 among older adults.

Corresponding author
Ping Yeap Loh,
lohpingyeap@gmail.com

## INTRODUCTION

Carpal tunnel syndrome (CTS) is one of the most common peripheral entrapment neuropathies. CTS causes peripheral nervous system changes with respect to morphological and functional characteristics such as a reduction in sensory nerve conduction velocity (*Mold et al., 2004*; *Verdu et al., 2000*). Both the motor and sensory symptoms of CTS have substantial effects on work performance and quality of life. Indeed, the health-related quality of life is reduced in CTS patients compared with the general population (*Mold et al., 2004*).

The *United States Bureau of Labor Statistics (2008)* estimated a 36.5% increase of workers who are aged 55–64 years from 2006 to 2016. Notably, the incidence of CTS is highest in the age groups of 50–54 years and 75–84 years (*Bland & Rudolfer, 2003*). The onset of CTS among older adults causes severe symptoms such as weakness and atrophy of the thenar muscles, pain and numbness over the palm and fingertips, as well as motor and sensory axon loss (*Blumenthal, Herskovitz & Verghese, 2006*). CTS has severe social and economic impacts such as a reduction in personal and industrial productivity, lower income among

**Table 1  Demographic data of participants ($n = 34$).**

|  |  | Mean ± SD |
|---|---|---|
| Age (years) |  | 70.9 ± 5.2 |
| Height (cm) |  | 164.9 ± 6.1 |
| Weight (kg) |  | 62.5 ± 10.1 |
| BMI (kg/m$^2$) |  | 22.9 ± 2.9 |
| Wrist Circumference (mm) | Right | 167.3 ± 9.7 |
|  | Left | 165.1 ± 9.2 |
| Handedness | Right hand dominant | 33 |
|  | Left hand dominant | 1 |

employees, and increase of medical treatment costs (*Foley, Silverstein & Polissar, 2007*). Therefore, it is important to create an age-friendly working environment to reduce the risk of CTS among the aging workforce.

The carpal tunnel volume becomes smaller as the wrist position changes into flexion or extension (*Mogk & Keir, 2009*). Changes of the wrist posture also cause deformation of the median nerve at the carpal tunnel. Wrist flexion-extension positions cause the median nerve cross-sectional area (MNCSA) to become smaller, whereas wrist radial-ulnar deviations do not have significant effects on the MNCSA (*Loh & Muraki, 2014*). *Loh & Muraki (2015)* reported that a higher wrist flexion/extension position causes increased deformation of the MNCSA compared with the neutral wrist position.

The aging process is known to cause physiological and anatomical changes to the hands as well as the peripheral nervous system (*Carmeli, Patish & Coleman, 2003*). *Loh & Muraki (2015)* reported changes in median nerve behavior at different wrist angles among young participants. However, the median nerve behaviors with respect to changes in the wrist angle among older adults are not well understood. Therefore, the objective of the present study was to investigate the age-related changes of median nerve behavior, namely MNCSA and the longitudinal (D1) and vertical (D2) diameters, at different wrist angles among older adults.

## MATERIALS AND METHODS

### Participants

This study was approved by the Ethics Committee of the Faculty of Design at Kyushu University (Approval number, 141; June 04, 2013). In a previous study (*Loh & Muraki, 2015*), there was no difference in deformation pattern in the MNCSA, D1, and D2 between the young female and male groups. Therefore, in this study, we recruited only male participants. Thirty-four healthy men were recruited and written informed consent was obtained (Table 1). The inclusion criteria for participation were that the subject was able and willing to provide informed consent and was above 60 years of age. The exclusion criteria for participation were patients with CTS, diabetes, or a history of wrist surgery or wrist fracture. All participants were free of signs and symptoms of CTS, as indicated by screening tests, including the Boston Carpal Tunnel Questionnaire, Phalen Test, and

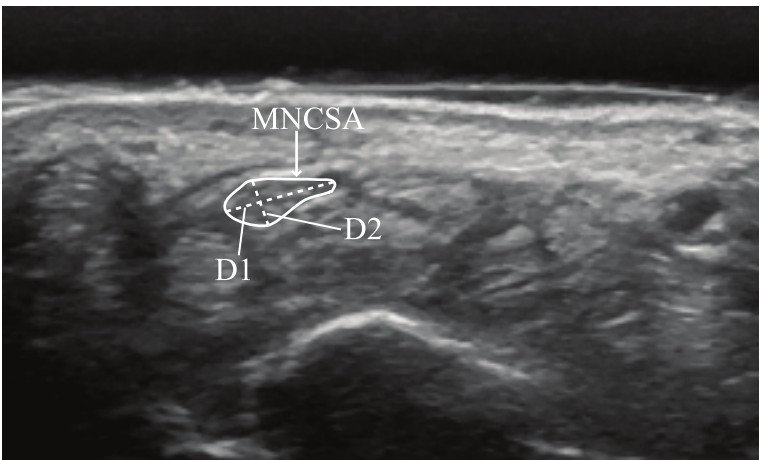

**Figure 1** Quantifying median nerve cross-sectional area (MNCSA), median nerve longitudinal diameter (D1), and median nerve vertical diameter (D2) at the proximal carpal tunnel.

CTS Tinel Test (*LaJoie et al., 2005*; *Sambandam et al., 2008*). The Edinburgh Handedness Inventory (*Oldfield, 1971*) was used to identify the handedness of the participants. Table 1 presents the demographic data of the participants.

## Ultrasound examination

The ultrasound examination protocol was conducted as described by *Loh & Muraki (2015)*. Ultrasound examination was conducted with the LOGIQ e ultrasound system (GE Healthcare, Little Chalfont, UK) and a 7.0-mm-thick sonar pad (Nippon BXI Inc., Tokyo, Japan) as a coupling medium. A total of seven wrist positions were examined as follows: neutral (0°); 15°, 30°, and 45° extension; and 15°, 30°, and 45° flexion. A wrist goniometer was used to determine the passive wrist angle during the ultrasound examination. The fulcrum of the goniometer was placed at the triquetrum, while the static arm was placed parallel to the ulnar bone and the moveable arm was placed parallel to the fifth metacarpal bone. Three images were taken for each wrist position during the ultrasound examination. Ultrasound examination with the passive wrist angle was repeated for both the dominant and nondominant hand.

The examiner palpated the pisiform bony mark and placed the ultrasound probe at the pisiform level to identify the proximal edge of the transverse carpal ligament and the median nerve in the transverse plane by a hypoechogenic rim, which contained hypoechogenic nerve fascicles. The MNCSA, D1, and D2 (Fig. 1) were quantified using ImageJ software (*Schneider, Rasband & Eliceiri, 2012*). The MNCSA was measured by the tracing method along the hypoechogenic rim of the median nerve; the D1 and D2 were measured as described by *Duncan, Sullivan & Lomas (1999)*. *Loh & Muraki (2015)* reported good to excellent inter- and intrarater reliability in quantification of the MNCSA and median nerve diameters. Mean values of three images were calculated to represent the MNCSA, D1, and D2 at each wrist angle.

**Table 2** Normality test for median nerve cross-sectional area ($n = 34$).

| Wrist | Skewness (M ± SE) | Kurtosis (M ± SE) | Shapiro–Wilk Test (*P* value) |
|---|---|---|---|
| Dominant | 0.46 ± 0.40 | −0.59 ± 0.79 | .233 |
| Nondominant | 0.52 ± 0.40 | −0.42 ± 0.79 | .188 |

**Notes.**
M, mean; SE, standard error.

## Deformation percentage

Deformation percentages of the MNCSA, D1, and D2 at different wrist angles relative to values in the neutral wrist position were calculated with the following equation:

$$\text{Deformation Percentage} = \frac{\text{Neutral wrist} - \text{Different wrist angle}}{\text{Neutral wrist}} \times 100\%.$$

## Statistical analysis

Statistical analysis was performed using SPSS version 21.0 software (IBM Corporation, Chicago, Illinois, USA). The sample characteristics of MNCSA in both the dominant and nondominant hand were examined by the Shapiro–Wilk's normality test.

The paired $t$-test was used to analyze differences in MNCSA, D1, and D2 between the dominant and nondominant hands at neutral position. Two-way repeated analysis of variance ($2 \times 7$ factorial design) was conducted with wrist side (dominant and nondominant) and the seven wrist angles (neutral [0°]; 15°, 30°, and 45° extension; and 15°, 30°, and 45° flexion) as factors to examine differences in MNCSA, D1, and D2 at different wrist positions. Greenhouse-Geisser correction was used in the analysis of variance because the assumption of sphericity was violated, as indicated by Mauchly's test. Post-hoc pairwise Bonferroni-corrected comparison was used to examine mean differences in the factors. The significance level was set at 0.05 (5%) for all statistical tests.

# RESULTS

## Sample characteristics

The Shapiro–Wilk's test ($P > .05$), and visual inspection of histograms, normal Q–Q plots, and box plots indicated the MNCSA values of both hands were approximately normally distributed and slightly skewed and kurtotic (Table 2) (*Cramer & Howitt, 2004*; *Doane & Seward, 2011*; *Razali & Wah, 2011*; *Shapiro & Wilk, 1965*).

## Comparison of median nerve behavior between the dominant and nondominant hands at neutral wrist position

The MNCSA and D1 values of the dominant hand were significantly larger ($P < .001$) than those of the nondominant hand, while D2 showed no significant difference between the hands ($P = .137$) (Table 3).

**Table 3 Comparison between dominant and nondominant hands at wrist neutral ($n = 34$).**

|  | Dominant hand | Nondominant hand | $t$ | $P$ |
|---|---|---|---|---|
| MNCSA ($mm^2$) | 9.65 ± 2.17 | 8.70 ± 1.88 | 7.186 | <.001 |
| D1 (mm) | 6.02 ± 0.91 | 5.58 ± 0.80 | 5.398 | <.001 |
| D2 (mm) | 2.01 ± 0.21 | 1.96 ± 0.25 | 1.524 | .137 |

Notes.
MNCSA, median nerve cross-sectional area; D1, median nerve longitudinal diameter; D2, median nerve vertical diameter.

**Table 4 Mean values of median nerve cross-sectional area, longitudinal diameter and vertical diameter at different wrist positions.**

| Wrist angle | MNCSA ($mm^2$) | | D1 (mm) | | D2 (mm) | |
|---|---|---|---|---|---|---|
|  | Dom | Nondom | Dom | Nondom | Dom | Nondom |
| Flexion 45° | 7.65 ± 1.79 | 6.92 ± 1.46 | 4.78 ± 1.04 | 4.44 ± 0.91 | 2.03 ± 0.24 | 1.98 ± 0.26 |
| Flexion 30° | 8.08 ± 2.04 | 7.39 ± 1.59 | 5.09 ± 1.08 | 4.81 ± 0.82 | 2.02 ± 0.26 | 1.95 ± 0.23 |
| Flexion 15° | 8.92 ± 2.09 | 8.10 ± 1.78 | 5.55 ± 1.07 | 5.21 ± 0.91 | 2.00 ± 0.20 | 1.96 ± 0.19 |
| Neutral (0°) | 9.65 ± 2.17 | 8.70 ± 1.88 | 6.02 ± 0.91 | 5.58 ± 0.80 | 2.01 ± 0.21 | 1.96 ± 0.25 |
| Extension 15° | 8.87 ± 1.95 | 8.02 ± 1.61 | 5.95 ± 0.74 | 5.59 ± 0.69 | 1.88 ± 0.23 | 1.80 ± 0.24 |
| Extension 30° | 8.09 ± 1.79 | 7.43 ± 1.53 | 5.81 ± 0.70 | 5.49 ± 0.74 | 1.73 ± 0.25 | 1.72 ± 0.26 |
| Extension 45° | 7.52 ± 1.62 | 6.91 ± 1.46 | 5.80 ± 0.73 | 5.47 ± 0.82 | 1.63 ± 0.23 | 1.59 ± 0.20 |

Notes.
MNCSA, median nerve cross-sectional area; D1, median nerve longitudinal diameter; D2, median nerve vertical diameter; Dom, dominant hand; Nondom, nondominant hand.

**Table 5 Deformation percentage of median nerve cross-sectional area (MNCSA) ($mm^2$) at difference wrist positions when compare to wrist neutral position.**

| MNCSA ($mm^2$) | Flex 45° | Flex 30° | Flex 15° | Neutral | Extn 15° | Extn 30° | Extn 45° |
|---|---|---|---|---|---|---|---|
| Dom | −20.3% | −16.3% | −8.4% | NA | −7.8% | −15.8% | −21.6% |
| NonDom | −19.8% | −14.4% | −7.8% | NA | −7.4% | −14.1% | −20.0% |

Notes.
Dom, dominant hand; NonDom, nondominant hand; Flex, flexion; Extn, extension; NA, Not applicable.

## Change in MNCSA at different wrist angles

No significant wrist angle × handedness interaction was found ($F[4.1, 135.9] = 1.135$, $P = .267$). However, wrist angle had a significant effect on MNCSA, which became smaller when the wrist changed from a neutral to a flexion or extension position in both the dominant and nondominant hands. The MNCSA at neutral position (0°) was significantly larger as compared with 15°, 30°, and 45° flexion and extension, respectively (Figs. 2A and 2B). The mean values and the deformation percentages of the MNCSA at different wrist positions are presented in Tables 4 and 5, respectively.

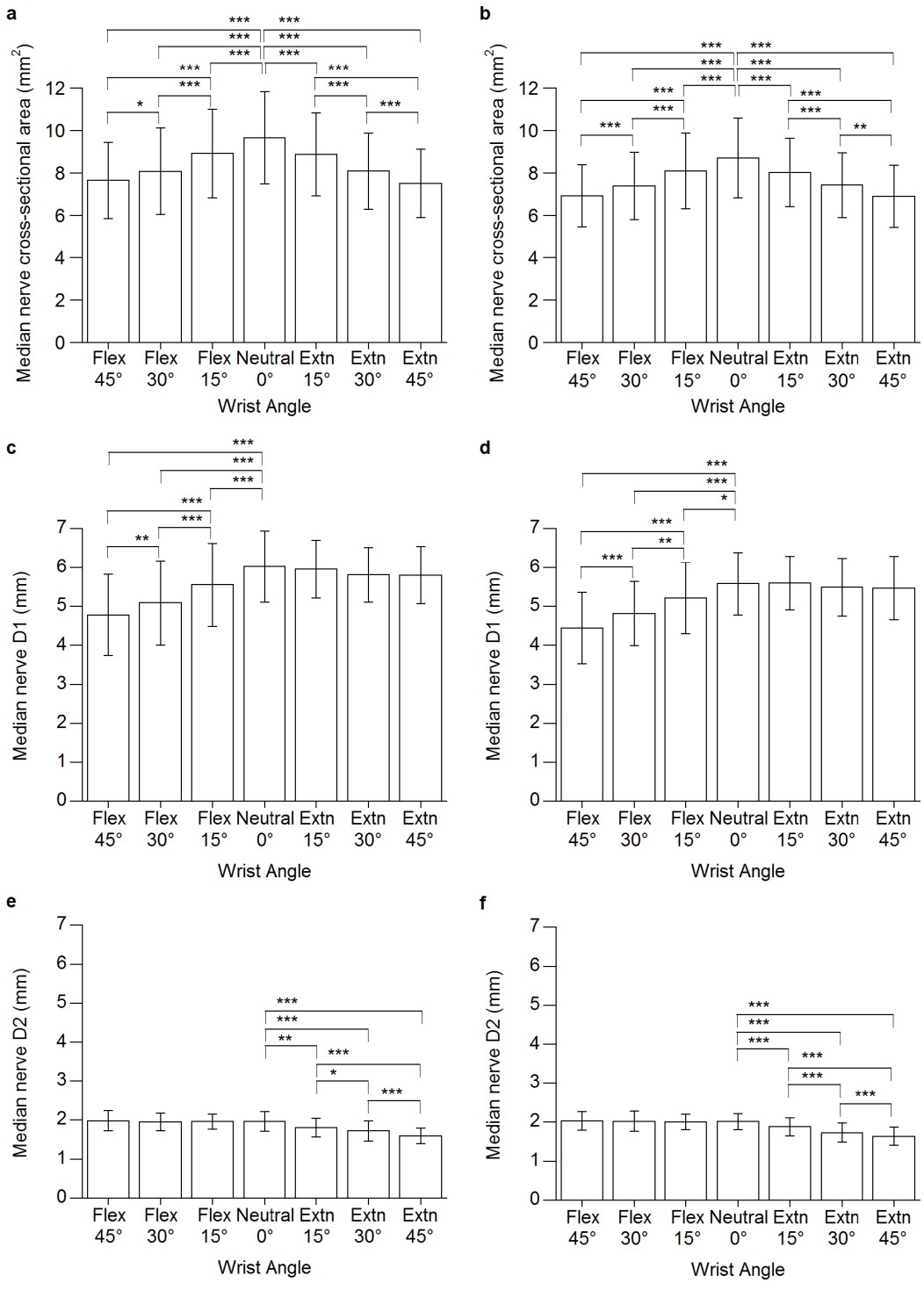

**Figure 2 Median nerve cross-sectional area at different wrist angles.** (A) Dominant hand. (B) Non-dominant hand; Median nerve longitudinal diameter (D1) at different wrist angles (C) Dominant hand. (d) Nondominant hand; Median nerve vertical diameter (D2) at different wrist angles (E) Dominant hand (F) Nondominant hand. Extn, extension; Flex, flexion. *$P < .05$, **$P < .01$, ***$P < .001$.

**Table 6 Deformation percentage of median nerve longitudinal diameter (D1) (mm) at difference wrist positions when compare to wrist neutral position.**

| D1 (mm) | Flex 45° | Flex 30° | Flex 15° | Neutral | Extn 15° | Extn 30° | Extn 45° |
|---|---|---|---|---|---|---|---|
| Dom | −20.6% | −15.6% | −9.8% | NA | −0.7% | −2.8% | −3.0% |
| NonDom | −20.4% | −13.6% | −8.2% | NA | 0.7% | −1.2% | −1.3% |

Notes.

Dom, dominant hand; NonDom, nondominant hand; Flex, flexion; Extn, extension; NA, Not applicable.

**Table 7 Deformation percentage of median nerve vertical diameter (D2) (mm) at difference wrist positions when compare to wrist neutral position.**

| D2 (mm) | Flex 45° | Flex 30° | Flex 15° | Neutral | Extn 15° | Extn 30° | Extn 45° |
|---|---|---|---|---|---|---|---|
| Dom | 1.5% | 0.9% | −0.9% | NA | −6.3% | −14.0% | −18.8% |
| NonDom | 1.4% | 0.3% | −0.5% | NA | −7.7% | −11.8% | −18.7% |

Notes.

Dom, dominant hand; NonDom, nondominant hand; Flex, flexion; Extn, extension; NA, Not applicable.

## Changes in D1 and D2 at different wrist angles

The mean D1 and D2 values at different wrist angles are presented in Table 5. The results showed no significant wrist angle × handedness interaction for both D1 and D2 (D1: $F[4.0, 130.5] = .399$, $P = .807$; D2: $F[3.6, 118.0] = .601$, $P = .644$). Wrist flexion had a significant influence on D1, which became smaller compared with that at neutral position (Figs. 2C and 2D). Meanwhile, wrist extension caused a significant decrease in D2 (Figs. 2E and 2F). The mean values and deformation percentages of the D1 and D2 at different wrist positions are presented in Table 4 and Tables 6 and 7, respectively.

## DISCUSSION

### Aging in the peripheral nerve

The aging process causes specific changes to the peripheral nerve such as a reduction in conduction velocity, as well as biomechanical and morphological changes (*Verdu et al., 2000*). The process of peripheral nerve degeneration is associated with the phenomenon of the loss of nerve fibers (*Kerasnoudis et al., 2013*). Furthermore, peripheral nerve morphological studies suggested that aging causes a reduction in the percentage of myelinated axons while increasing the epineural and fascicular areas, with a larger neural cross-sectional area observed in the elderly group (*Kerasnoudis et al., 2013*; *Kundalić et al., 2014*). The mean MNCSA value in young adults and the elderly group (>60 years old) were 8.75 mm$^2$ and 9.12 mm$^2$, respectively (*Kerasnoudis et al., 2013*). The mean values of the MNCSA at the neutral wrist position in the older adults (Table 4) were larger compared with those of the young adults (Dominant = $8.36 \pm 1.47$ mm$^2$; Nondominant = $7.32 \pm 1.40$ mm$^2$) reported in our previous study (*Loh & Muraki, 2015*). Furthermore, the D1 values of the older adults were generally larger than those of the

young adults, which was not the case for the D2 values. Therefore, the longer D1 may be the main factor contributing to the larger MNCSA observed among older adults.

## Deformation of the MNCSA among older adults

A larger angle in wrist flexion and extension led to increased deformation of the MNCSA at the proximal carpal tunnel level (Table 5). The carpal tunnel volume decreases as the wrist moves to extension and flexion positions, which leads to increased intra-carpal tunnel pressure, resulting in median nerve compression from the surrounding tendons within the carpal tunnel (*Mogk & Keir, 2009*; *Vital et al., 1990*). The CSA of a nerve is reduced when it undergoes elongation, in a process known as transverse contraction (*Keir & Rempel, 2005*). The epineurium layer helps to constrain the inner core pressure of the nerve and contributes to the stiffness of the nerve in resisting the transverse contraction (*Keir & Rempel, 2005*; *Millesi, Zoch & Reihsner, 1995*).

Morphometric studies of the peripheral nerve have indicated the quantitative and qualitative changes that occur during the aging process, such as an increased CSA and thickening of the peripheral nerve sheath of a peripheral nerve (*Kundalić et al., 2014*; *Topp & Boyd, 2006*). The thicker epineurium in older adults, which may explain their larger MNCSA, may affect the biomechanical properties of a nerve owing to the stress imposed on the nerve during joint movement. Similar to the observations in young adults (*Loh & Muraki, 2015*), an increase of 15° in wrist flexion or extension caused a significant reduction and higher deformation percentage of the MNCSA (Figs. 2A and 2B). However, the deformation of MNCSA at 45° flexion and extension was reduced in older adults (−20%) compared with young adults (−25%). Therefore, the thicker peripheral nerve sheath among older adults may help to provide stronger inner pressure of a nerve while responding to the stretch and compression stress occurring during wrist angle changes.

## Deformation of the D1 and D2 among older adults

At a neutral wrist position, the D1 values of older adults (Table 4) were larger than those of the young group of our previous study (*Loh & Muraki, 2015*) (Dominant = 5.09 ± 0.59 mm; Nondominant = 4.79 ± 0.66 mm), whereas the D2 values were similar between older adults (Table 4) and young adults (Dominant = 2.11 ± 0.23 mm; Nondominant = 1.95 ± 0.26 mm). Furthermore, the deformation pattern of D1 at wrist flexion and D2 at extension positions are similar to those of the young adults as reported by *Loh & Muraki (2015)*. The compression stress and elongation at wrist flexion-extension caused the D1 to become smaller as the wrist position changed into flexion positions, whereas the D2 became shorter at the wrist extension positions (Figs. 2C–2F). The loose space within the epineural tube enables a change of the fascicular position so that the peripheral nerve can adapt to the biomechanical stress such as strain, excursion, and transverse contraction experienced during limb movement (*Keir & Rempel, 2005*; *Sladjana, Ivan & Bratislav, 2008*; *Topp & Boyd, 2012*). Therefore, the nerve behavior at different joint positions of older adults can be observed via changes in the nerve diameters.

The carpal tunnel is a confined space and is tightly packed with flexor tendons and a median nerve. A larger D1 and MNCSA may indicate that the median nerve occupies

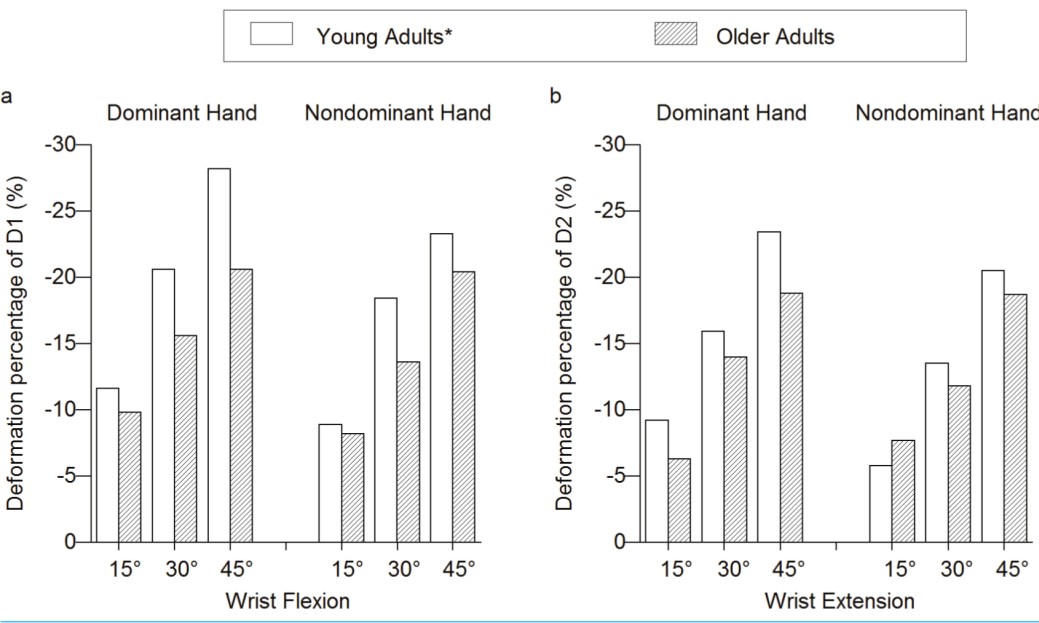

**Figure 3 Deformation percentages between young and older adults.** (A) Longitudinal diameter, D1 (B) Vertical diameter, D2. * Cited from *Loh & Muraki (2015)*

relatively more space at the carpal tunnel. Therefore, the higher deformation percentages of D1 and D2 at different wrist positions among older adults could reflect their larger MNCSA at the proximal carpal tunnel level. However, the lower deformation observed among older adults (Figs. 3A and 3B) suggested age-related changes of median nerve behavior at different wrist angles. The lower deformation percentages among older adults may indicate a thickened median nerve sheath and lack of movement among the polyfascicular tube within the epineural tube. Undulation mechanisms play an important role when a nerve is exposed to the elongation and shortening stress across a joint or multijoint, because the loose arrangement of the perineural tube within the epineurium sheath allows for volume adaptation of a nerve (*Keir & Rempel, 2005*; *Sladjana, Ivan & Bratislav, 2008*). When a joint is flexed, the nerve that spans across the joint becomes shorter by increasing the number of perineurium undulations. In contrast, when the joint is extended, the folded undulations will be straightened in response to the longitudinal elongation stress (*Keir & Rempel, 2005*). Therefore, undulation mechanisms may cause the nerve diameters to change accordingly during joint movement. Furthermore, the physical connection between the core and sheath of the nerve, and the total fascicle number within the nerve are known to affect the stiffness of a nerve (*Tillett et al., 2004*). The increase in the ratio between the perineural thickness of the sural nerve and the fascicular diameter may increase the risk of chronic trauma over the nerve (*Tohgi, Tsukagoshi & Toyokura, 1977*). Therefore, aging nerves are more vulnerable to stress.

The aging process causes degenerative changes of wrist-carpal and carpal joints, which leads to decreased mobility of wrist movements and affects older adults in their occupation and daily life activities (*Carmeli, Patish & Coleman, 2003*). The diameters of the carpal tunnel may increase with the aging process, as the carpal tunnel volume of older adults

are larger than those of younger adults (*Pierre-Jerome, Bekkelund & Nordstrøm, 1997*). Our results suggest that the MNCSA of the median nerve is larger among older adults, with a longer D1, compared with younger adults. The median nerve undergoes shear strain and compression stress owing to the excursion and displacement of tendons within the confined space of the carpal tunnel (*Van Doesburg et al., 2010*). The long-term effects of transverse contraction force, shear strain, and compression stress within the narrow carpal tunnel may cause the appearance of the median nerve to change with the aging process.

### Aging workforce and CTS

Age-related anatomical and physiological changes have strong correlations with decreased functions in the central nervous system as well as the peripheral nervous system. Changes in the peripheral nervous system among older adults affects the somatosensory system, physical work performance, and quality of life (*Carmeli, Patish & Coleman, 2003*; *Sachs, 2008*). *Blumenthal, Herskovitz & Verghese (2006)* hypothesized that older adults underreport their CTS symptoms, which results in severe median nerve entrapment compared with young adults, as indicated in muscle wasting and nerve conduction studies.

Humans have distinct hand functions such as a powerful grip, precision grip, and precise thumb and finger movements. The normal aging process leads to a certain degree of deterioration in hand functions such as a decrease in manual dexterity and loss of hand strength, which in turn affects the strength, motor skills, work tolerance level, and speed of movement. In view of the increasing population of the aging workforce and the implication of age-related problems of the upper extremities, specific ergonomics considerations pertaining to the wrist angle in the design of tools and equipment and in the workplace are required to minimize the compression stress on the median nerve.

## ACKNOWLEDGEMENTS

The authors thank all of the individuals who participated in this study.

### Funding

The authors received no specific funding for this study.

### Competing Interests

The authors declare there are no competing interests.

### Author Contributions

- Ping Yeap Loh conceived and designed the experiments, performed the experiments, analyzed the data, contributed reagents/materials/analysis tools, wrote the paper, prepared figures and/or tables.
- Hiroki Nakashima performed the experiments, analyzed the data, contributed reagents/materials/analysis tools, prepared figures and/or tables, reviewed drafts of the paper.

- Satoshi Muraki conceived and designed the experiments, contributed reagents/materials/analysis tools, reviewed drafts of the paper.

## Human Ethics

The following information was supplied relating to ethical approvals (i.e., approving body and any reference numbers):

1. Ethics Committee of the Faculty of Design at Kyushu University
2. Approval number, 141 (June 04, 2013).

## Supplemental Information

Supplemental information for this article can be found online at http://dx.doi.org/10.7717/peerj.928#supplemental-information.

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
