# Peer review of "Median nerve behavior at different wrist positions among older males"

_PeerJ, doi:10.7717/peerj.928_

## Round 0.1 · original submission · Minor Revisions

Dear Authors,

Please edit the manuscript as per suggestions of the first and third peer reviewer and return the manuscript back to PeerJ as soon it is done.

·

Basic reporting

NO comments

Experimental design

1. Any objective landmark were defined on the ultrasound, so that you can define the same location across different subjects?
2. How to define the wrist angle precisely? the intervals are very close.
3. Why using paired t test not student t-test? (not repeating the same set of experiment)
4. The measurement at the carpal tunnel inlet is nearly impossible when the wrist was flexed at 30 to 45 degrees, the ultrasound probe can not put there. I doubt the level of carpal tunnel inlet of your measurement were changing all the time during the study. Also, you may need to tilt the probe and thus the true CSA may not be obtained.
5. Fig.1, why define D2 as the midpoint intersect of D1, but not the longest distance on the Y axis? That will reduce the D2 value and give us an impression the nerve is much compressed.
6. The major clinical problem is the nerve being compressed and flatten, why there was no corresponding analysis on D1/D2 ratio? individual D1 and D2 values have no clinical meaning but structural information.

Validity of the findings

1. The clinical importance is the increase carpal pressure that affect the morphology of the median nerve. If the pressure point at various wrist angle is exactly the same as the US scanning plane, then the decrease in the CSA is expected. However, I am not sure the authors is actually measuring this. Otherwise, in another similar study, the MN CSA away from the pressure point was measured as increased not reduced.
2. At least the most persuasive finding is that the MN has become swollen with age, but since the comparison were from different studies (different methodology?), precaution should be taken when appreciating the result.
3. Please elaborate the observed difference between the dominate and non-dominate hand? To me, the non-dominate hand also at risk, especially when I use my mobile phone (I hold the phone using non-dominant hand at an improper wrist angle- extension and for a substantial period)

As a whole, further information about the level of US is needed in order to confirm the results generated from this study is correct or not.( the authors claimed that they are measuring within carpal tunnel not the outside.

Additional comments

It would be great if they have an author having medical background.

Reviewer 2 ·

Basic reporting

NO COMMENTS

Experimental design

NO COMMENTS

Validity of the findings

NO COMMENTS

·

Basic reporting

This study is an interesting research and well-designed study. I think that this study is useful in the clinic. But I hope to suggest one thing. It is good to delete Table 8.
I think that only to make mention of the numeric value of the results acquired in the your previous study is enough in Discussion. Table 8 has a risk to give us misunderstanding. So, I recommend you to delete table 8.

Experimental design

This study is a well-designed study.

Validity of the findings

Validity is no problem.

Additional comments

For the additional study about this issue, I recommend you to research the relationship between the wrist joint angle and the speed of median nerve conduction, and the relationship between the thickness of median nerve and the speed of median nerve conduction. At that time, we can understand fully the changes of median nerve function according to changes of wrist angle.
Thank you for submitting your good paper.

---

## Round 0.2 · accepted · Accept

The revised manuscript has been accepted for publication and will undergo futher processing according to Peer J's format

·

Basic reporting

No comments

Experimental design

No comments

Validity of the findings

No comments

Additional comments

Thank you for your responds. Your reply is concise and accurate.
The revised version looks fine, and I have no more additional comments.
Good luck and all the best.